# Impact of the Allergic Therapeutic Adherence in Children with Allergic Rhinitis and ADHD: A Pilot Study

**DOI:** 10.3390/jpm13091346

**Published:** 2023-08-31

**Authors:** Antonella Gambadauro, Simone Foti Randazzese, Arianna Currò, Francesca Galletta, Giuseppe Crisafulli, Lucia Caminiti, Eva Germanò, Gabriella Di Rosa, Antonio Gennaro Nicotera, Sara Manti

**Affiliations:** 1Pediatric Unit, Department of Human Pathology in Adult and Developmental Age “Gaetano Barresi”, University of Messina, 98124 Messina, Italy; gambadauroa92@gmail.com (A.G.); simone.foti.92@gmail.com (S.F.R.); francygall.92@gmail.com (F.G.); crisafullig@unime.it (G.C.); lucycaminiti@yahoo.it (L.C.); saramanti@hotmail.it (S.M.); 2Unit of Child Neurology and Psychiatry, Department of Human Pathology in Adult and Developmental Age “Gaetano Barresi”, University of Messina, 98124 Messina, Italy; arianna.curro0511@gmail.com (A.C.); eva.germano@unime.it (E.G.); gdirosa@unime.it (G.D.R.)

**Keywords:** allergic rhinitis, attention deficit hyperactivity disorder, ADHD, therapeutic adherence, children

## Abstract

Background: Allergic rhinitis (AR) is the most common chronic allergic disease in children. Several studies have shown an association between attention deficit hyperactivity disorder (ADHD) and allergies, especially AR. Patients with ADHD usually have poor therapeutic adherence, and untreated AR symptoms may worsen the quality of life of patients. Methods: The aim of our study was to analyse therapeutic adherence in patients with ADHD and AR and estimate the impact of the adherence on ADHD symptoms. Total Nasal Symptoms Score (TNSS), Paediatric or Adolescent Rhinoconjunctivitis Quality of Life Questionnaire (PRQLQ 6–12 years; ARQLQ 13–17 years), Swanson, Nolan, and Pelham version IV scale (SNAP-IV), and Medication Assessment Questionnaire (MGL MAQ) were recorded. Results: In the AR-ADHD group, a positive correlation between TNSS and SNAP-IV subscales was found: worse AR symptoms were related to a negative effect on ADHD scores. AR-ADHD patients with better ADHD therapeutic adherence showed higher AR symptoms and higher oppositional defiant disorder scores in the SNAP-IV questionnaire. Conclusions: Our results suggest that better adherence to AR therapy (oral antihistamines and/or intranasal corticosteroids, INCS) is associated with a reduction in inattention symptoms in children with ADHD. This data could prove to be fundamental for the psychic outcome of these patients.

## 1. Introduction

Allergic rhinitis (AR) is the most common chronic allergic disease in school-age children [1,2], with an estimated global prevalence of 40% [3,4]. AR is clinically characterized by nasal obstruction, itching, sneezing, and rhinorrhoea. Objective tests for diagnosis of IgE-mediated forms are the skin prick test (SPT) and serum-specific IgE levels [5]. Children with AR may also exhibit hyperactive behaviour, sleep disturbances, and poor school performance, which has been thought to be a consequence of the chronic and disturbing illness [6].

Attention deficit hyperactivity disorder (ADHD) is a common neurobehavioral disorder affecting 7.2% of children worldwide [7]. Characteristic symptoms of ADHD are inattention, impulsivity, and hyperactivity that often lead to social impairments and decreased academic performance [8]; furthermore, sleep disturbances have also been described in these patients [9]. ADHD may be associated with a number of comorbid psychiatric disorders, including oppositional defiant disorder (ODD), conduct disorder, learning disabilities, substance use disorder, and mood and anxiety disorders. Moreover, ADHD is associated with multifarious negative outcomes, which may be exacerbated by the presence of comorbid disorders [10]. The diagnosis of ADHD is based on Diagnostic and Statistical Manual of Mental Disorders, Fifth Edition (DSM-5) criteria [11].

A recent systematic review and meta-analyses [12] have shown an association between ADHD and allergies, including AR. In fact, patients with allergic diseases have a 30–50% greater risk of developing ADHD. A large population-based case–control study conducted in Taiwan showed an increased prevalence of allergic diseases among ADHD patients compared to the healthy controls, including AR (OR = 1.59) [13]. A longitudinal study enrolling 549 Korean children with ADHD showed that the relative risk of AR was 1.38 times greater compared to controls without ADHD [14]. It has been suggested that the positive correlation between allergic diseases and behavioural disorders may be related to a common biological background since immune dysregulation and inflammation play a central role in the development of both conditions [15,16].

As underlined above, both AR and ADHD have a negative impact on social habits, school performance, and the sleep quality of affected children, especially in untreated patients.

The purpose of our study was to analyse the therapeutic adherence in a small group of children with ADHD and AR, while also evaluating if satisfactory medication adherence may improve symptoms control and the quality of life of patients suffering from AR and ADHD.

## 2. Materials and Methods

### 2.1. Study Design

#### 2.1.1. Type of Study

Pilot, prospective, monocentre case–control study.

#### 2.1.2. Subjects and Eligibility Criteria: Case Group

Children of both genders, aged 6–18 years old, with a confirmed diagnosis of AR [5] and ADHD [11], who assumed at least one medication for ADHD (i.e., methylphenidate and/or risperidone) and AR (i.e., oral antihistamines and intranasal corticosteroids, INCS) and who were referred to Paediatric Unit and Paediatric Neuropsychiatry Department, University of Messina, Italy, were enrolled in the study from March 2023 to May 2023.

Exclusion criteria included: children younger than 6 years old and older than 18 years old without a diagnosis of ADHD and AR.

#### 2.1.3. Subjects and Eligibility Criteria: Control Group

Age-matched children with AR and requiring medications for AR, without ADHD, who had a regular follow-up in the Paediatric Unit, University of Messina, Italy, were enrolled as the control group. These children fulfilled criteria for AR diagnosis according with the ARIA guidelines [5].

Exclusion criteria included: children younger than 6 years old and older than 18 years old without a diagnosis of AR and without an intermittent or chronic therapy for AR (i.e., oral antihistamines and intranasal corticosteroids, INCS).

### 2.2. Study Procedures

Parents of all enrolled children completed validated questionnaires about the AR symptoms (Total Nasal Symptoms Score, TNSS) [17], the quality of life (Paediatric or Adolescent Rhinoconjunctivitis Quality of Life Questionnaire, PRQLQ 6–12 years—ARQLQ 13–17 years) [18,19], the ADHD symptoms (Swanson, Nolan, and Pelham version IV scale, SNAP-IV) [20], and the therapeutic adherence (4-item Morisky Green Levine Medication Assessment Questionnaire, MGL MAQ) [21]. All patients were analysed for therapeutic adherence to specific AR medication, i.e., oral antihistamines (cetirizine, levocetirizine, or bilastine) and INCS (mometasone or budesonide nasal spray) [22], while only ADHD patients were evaluated about therapeutic adherence for specific ADHD medications, such as methylphenidate and/or risperidone [23,24].

#### 2.2.1. AR Symptoms Score

The Total Nasal Symptom Score (TNSS) [17] was used to evaluate the AR severity in the two groups of patients, quantifying nasal obstruction, sneezing/nasal itching, and rhinorrhoea. All these symptoms were graded on a four-point scale (0, none; 1, mild; 2, moderate; 3, severe), and the scores were summed to have the total with a higher score equalling worse AR symptoms.

#### 2.2.2. Quality of Life Questionnaire

The impact of AR symptoms on the quality of life was analysed in the two groups through two different questionnaires: the Paediatric Rhinoconjunctivitis Quality of Life Questionnaire (PRQLQ), approved for children 6–12 years of age; and the Adolescent Rhinoconjunctivitis Quality of Life Questionnaire (ARQLQ) approved for adolescents 13–17 years of age [18,19].

The PRQLQ consists of 23 questions about five items: nose symptoms, eyes symptoms, practical problems, other symptoms, and activity limitations. The score of each question ranges from 0 (no impairment) to 6 (maximum impairment); herein, higher scores are related to a worse quality of life.

The ARQLQ is a 25-item questionnaire that adds a sixth item (emotions) to the other five. The score ranges are the same the PRQLQ.

#### 2.2.3. ADHD Symptoms

The Swanson, Nolan, and Pelham version IV scale (SNAP-IV) [20] is a 26-item questionnaire used to evaluate the severity of ADHD symptoms. It is divided into three subscales: inattention (9 questions), hyperactivity/impulsivity (9 questions), and oppositional defiant disorder (ODD, 8 questions). Each item is scored on a scale from 0 to 3, with higher scores related to more symptomatic patients.

#### 2.2.4. Therapeutic Adherence

The Morisky Green Levine Medication Assessment Questionnaire (MGL MAQ) [21] defined the two groups’ therapeutic adherence to AR medication. In contrast, only the AR-ADHD group was analysed for the therapeutic adherence to ADHD medication. The MGL MAQ is a 4-item questionnaire in which participants are asked about their experience in taking medicine. The answers are in a “yes” or “no” format, and each item is scored 0 (poor adherence) or 1 (good adherence).

### 2.3. Data Analysis

Data were collected and statistically analysed using Excel version 2013. Descriptive statistics (median and range) were used to describe demographic characteristics and clinical variables. The independent t-test was used to compare differences in age, TNSS, RQLQ, SNAPIV, and MGL MAQ scores between the AR-ADHD group and the controls. Fisher’s exact test was used to compare differences in preterm birth, exposure to parental tobacco smoke, neurological and allergic co-morbidities, sensitizing allergens, and different type of AR therapy (oral antihistamines, INCS or both combination) between the two groups. The relationship between possible explanatory parameters (age, severity of AR symptoms evaluated by TNSS, and medication adherence estimated by MGL MAQ) and the subscales of SNAP-IV were evaluated via Pearson correlation in the AR group and the AR-ADHD group. The statistical significance was set at the level of *p* < 0.05.

### 2.4. Ethics

The study was conducted in accordance with the Declaration of Helsinki, and all the participants’ parents gave written informed consent. This prospective observational study was carried out after the approval of the local Ethics Committee.

## 3. Results

A total of six AR-ADHD children and six AR control subjects were included in the study [Table 1 and Table 2]. All of the enrolled patients were male. The median age was 14 years for the AR-ADHD group and 13.5 for the AR group. No differences between the two groups based on age, preterm birth, exposure to parental tobacco smoke, sensitizing allergens, neurological and allergic co-morbidities, and quality of life (PRQLQ and ARQLQ) were recorded. Four out of six patients (66.6%) in the AR-ADHD group had neurological co-morbidities (i.e., tics and anxiety) compared to one out of six patients (16.6%) in the AR group who was experiencing neuro-psychiatric co-morbidities (i.e., tics and dyslexia).

With regard to allergic co-morbidities, 4/6 children (66.6%) in the AR-ADHD group showed comorbidities: 2/4 patients reported the diagnosis of asthma, and 3/4 patients were affected by allergic conjunctivitis. Among the AR group, 5/6 children (83.3%) had allergic diseases: four patients showed allergic asthma, two patients suffered from atopic dermatitis, and three patients reported a food allergy.

Patients of both groups were divided into three different AR medication categories: the first one included children on oral antihistamines therapy; the second comprised patients on INCS; the third included children on both therapies (oral antihistamines and INCS). Overall, 50% of patients in the AR-ADHD group were under oral antihistamines therapy; in the AR group, 100% of children were under a combination therapy (oral antihistamines and INCS), principally due to the severity of the AR symptoms at the onset and the presence of atopic comorbidities. Regarding the questionnaires, no significant differences in AR symptom scores (TNSS) and AR medication adherence (MGL MAQ) between the two groups (*p* = 0.58 and *p* = 0.26, respectively) were detected. However, children in the AR-ADHD group achieved slightly higher scores in TNSS than the controls (AR-ADHD group median value: 6.5 vs. AR-group median value: 6). Conversely, the AR group achieved higher scores in AR medication adherence (MGL MAQ) compared to the AR-ADHD group (AR group median value of 4 vs. AR-ADHD group median value of 2).

The six children of the AR-ADHD group demonstrated significantly higher scores in SNAP-IV compared to the controls (median value: 47 vs. 2.5, respectively; *p* = 0.008) [Table 3, Table 4 and Table 5].

We compared possible explanatory parameters (i.e., age, AR symptoms by TNSS, therapeutic adherence for AR and ADHD medications evaluated by MGL MAQ) with each subscale of the SNAP-IV in the AR-ADHD group [Table 6]. We identified a positive correlation between age and inattention subscale (r = 0.78; *p* = 0.06), TNSS and ODD subscale (r = 0.78; *p* = 0.06), and MAQ ADHD therapy and ODD subscale (r = 0.88; *p* = 0.02). Although these data were not statistically significant, a negative correlation was found between age and hyperactivity/impulsivity subscale and between MAQ AR therapy and inattention subscale (*p* = 0.74 and *p* = 0.95, respectively).

We compared possible explanatory parameters (i.e., age, AR symptoms by TNSS, therapeutic adherence for AR medications evaluated by MGL MAQ) with each subscale of the SNAP-IV in the AR group [Table 7]. We did not identify a significant correlation between age and TNSS, while a strong negative correlation between each subscale of the SNAP-IV and MGL MAQ scores for AR medications was detected (*p* = 0.00018, *p* = 0.00002, and *p* = 0.00007 for the inattention, hyperactivity/impulsivity, and ODD subscales, respectively).

In the AR-ADHD group, we divided patients in three AR therapy subgroups (oral antihistamines, INCS, and combination therapy) and analysed the different impact on TNSS, SNAP-IV, and MAQ AR therapy scores [Table 8]. No significant differences were found in the three AR therapy subgroups (*p* = 0.21, *p* = 0.55, and *p* = 0.29 for the TNSS, SNAP-IV, and MAQ AR therapy questionnaires, respectively); however, the patients on combination therapy (oral antihistamines and INCS) totalized the higher mean scores in all the questionnaires compared to the other subgroups. This result may show that patients who need AR combination therapy are the ones with worse AR and ADHD symptoms.

Furthermore, a significant correlation was found between TNSS and number of sensitizing allergens in both groups of patients (*p* = 0.01 in AR-ADHD group vs. *p* = 0.00051 in the AR group) [Table 9].

## 4. Discussion

ADHD is a common neurodevelopmental disorder which is accompanied by several psychiatric comorbid conditions. The high rate of comorbidities could make it complicated and difficult to manage [25]. Several studies have shown that the presence of psychiatric comorbidities can worsen the prognosis/outcome, particularly in patients who do not undergo treatment (pharmacological and/or rehabilitation) [26,27].

In a Taiwan population-based study [28], a higher risk of developing ADHD in patients with allergic disorders was found in the ages group 6–11 years old (OR = 2.10; 95% CI = 1.85–2.39); furthermore, males had a higher risk of developing ADHD (OR = 3.76; 95% CI =3.26–4.32). Suwan et al. [6] investigated the prevalence of allergic diseases in ADHD children (*n* = 40) compared to non-ADHD controls (*n* = 40); in this study, they identified a higher number of maternal smoking in ADHD patients (3/40) but without a statistical significance (*p* = 0.241) and without specifying if the patients were allergic or not. Second-hand smoke exposure at home was defined as a potential risk factor for developing ADHD and allergic diseases (especially food allergy) in the study of Wong et al. [29].

In our study, we aimed to perform the following: (1) analyse the impact of demographic and social background (i.e., age, pre-term birth, exposure to parental smoking) in AR-ADHD versus AR children; (2) evaluate the influence of neurological and/or allergic co-morbidities, AR symptoms (TNSS), ADHD symptoms (SNAP-IV), quality of life (PRQLQ and ARQLQ), AR and/or ADHD medication adherence (MGL MAQ) in both groups of patients; (3) investigate the relationship between possible explanatory parameters (particularly AR and/or ADHD medication adherence via 4-item MGL MAQ) and the subscales of SNAP-IV.

Our study found no difference in age, pre-term birth, exposure to parental smoking, and neurological and/or allergic co-morbidities between both groups.

In our analysis, patients in the AR-ADHD group achieved slightly higher scores in AR symptoms (TNSS) compared to the controls, but without a significant difference (*p* = 0.58). Based on the Pearson correlation results, it was observed that in the AR-ADHD group, there was a positive correlation between TNSS and SNAP-IV subscales, particularly the hyperactivity/impulsivity subscale (r = 0.61; *p* = 0.19) and the ODD subscale (r = 0.78; *p* = 0.06). These data showed how higher scores on TNSS were related to a negative effect on ADHD scores in the AR-ADHD group. Instead, no association was found between SNAP-IV subgroups and TNSS scores in the controls. The quality of life scores (PRQLQ and ARQLQ) did not show a statistical correlation in both groups (*p* = 0.52 and *p* = 0.38, respectively), probably as a consequence of the small number of the sample. These results were in accordance with other studies. In particular, Feng et al. [30] found significant positive correlations between severe AR symptoms (TNSS) and higher ADHD scores compared to controls; in their cross-sectional study, Chen et al. [31] demonstrated that children with AR and ADHD had more severe nasal symptoms (evaluated by TNSS) than those without ADHD.

Previously, few studies have investigated the relationship between therapeutic adherence and AR symptoms in children with ADHD or with hyperactive/inattention symptoms. In 2014, Kim et al. [32] evaluated 797 AR children and 239 non-allergic rhinitis (NAR) children, comparing the mean attention score of CAT (Comprehensive Attention Test) at baseline and after 1 year: at baseline, CAT scores were significantly lower in the AR group than in the controls; after 1 year of treatment, children with AR showed improvement in attention. A prospective follow-up study [33] on 68 children with a new AR diagnosis and who were drug naïve demonstrated how TNSS and SNAP-IV scores decreased significantly (*p* < 0.001) after starting treatment for AR. Another recent 3-month prospective study [34] enrolled 81 children with chronic rhinitis (61 patients with AR and 22 patients with NAR): Vanderbilt ADHD Diagnostic Rating Scale (VADRS) scores decreased when compared with those at baseline (*p* = 0.005), with a significant decrease only in the AR group after treatment (*p* < 0.001).

Our study evaluated the relationship between therapeutic adherence (MGL MAQ) and the SNAP-IV subscales through the Pearson correlation in the AR-ADHD and AR patients. By analysing the AR therapeutic adherence in the AR-ADHD group, we found a slightly negative correlation only in the inattention subscale (r = −0.03; *p* = 0.95), while the other subscales showed a non-significant positive correlation. Conversely, when evaluating the ADHD therapeutic adherence in the AR-ADHD group, we detected a significant positive correlation (r = 0.87; *p* = 0.02) in the ODD subscale. In other words, contrary to expectations, patients with better ADHD therapeutic adherence showed higher AR symptoms (worse TNSS scores) and higher ODD scores in the SNAP-IV questionnaire. However, only 33.3% of AR-ADHD patients were under AR combination therapy, in contrast to the control group (100% in combination therapy). Data analysis showed that AR-ADHD patients with higher TNSS scores are those on dual therapy (INCS + oral antihistamines) and with a more significant therapeutic adherence to ADHD and AR medications [Table 8]. Data analysis showed that AR-ADHD patients with higher TNSS scores are those on dual therapy (INCS + oral antihistamines) and with a more significant therapeutic adherence to ADHD and AR medications [Table 8]. Due to the study’s small sample, it is difficult to draw conclusions on the matter. One possible hypothesis could be that more AR-ADHD patients need combination therapy to better control the AR symptoms. Furthermore, we did not record TNSS at the onset of AR symptoms in our patients, so we cannot exclude the possibility that AR symptoms were worse at the diagnosis, and the decision to administer a combination therapy was consequential to the severity of the symptoms. It is also worth noting that all the patients in the AR-ADHD group had dust mite allergy (the principal cause of perennial allergic rhinitis) as opposed to only 50% in the AR group, in which a SPT resulted negative for animal dander and moulds. When evaluating the AR group, we found a significant negative correlation between AR therapeutic adherence and all three SNAP-IV subscales: in this group, all the low scores in SNAP-IV questionnaire were in accordance with the high scores in AR therapeutic adherence. Regardless of the underlying cause and the psychiatric therapy taken by the patients, our data suggested that better AR therapeutic adherence in children could improve the inattention symptoms in ADHD patients. These considerations can strongly impact the prognosis of psychiatric problems of these children and suggest how important it is to take overall care of these patients.

## 5. Conclusions

The possible explanations of our results compared to other previous studies cited above should be interpreted in light of our study’s strengths and limitations. To our knowledge, this is the first study to analyse the two distinct paediatric populations separately from the start: the AR-ADHD group and the AR group. In fact, the previous studies usually evaluated AR children, identifying the possible ADHD group during the analysis based on the SNAP-IV scores. Furthermore, all our patients have a previous diagnosis (at least one year before) of AR and/or ADHD.

In our study, the patients in AR-ADHD group with worse symptoms seem to have a good AR therapeutic adherence, without a significant impact on their quality of life and on their symptoms. However, our results suggest that better adherence to AR therapy is associated with a reduction in inattention symptoms in children with ADHD. The data, if confirmed by further studies, could prove to be fundamental for the psychic outcome of these patients. The principal limitation of our study is the small sample size. However, this research was designed like a pilot study with the purpose of increasing the number of participants. In this perspective, a multicentre study maybe proposed with the possibility of increasing the number of possible explanatory variables (i.e., correlation with different types of AR/ADHD medication).

## Figures and Tables

**Table 1 jpm-13-01346-t001:** Characteristics of the AR-ADHD group.

Characteristics	Participants					
	1	2	3	4	5	6
Age	15	13	12	17	15	11
Natal preterm	−	−	−	+	−	−
Exposure to parental smoking	+	+	−	−	+	−
Neurological co-morbidities	+	+	−	−	+	+
Atopic co-morbidities	+	+	−	+	−	+
Positive SPT:						
House dust mites	+	+	+	+	+	+
Mold	−	−	−	+	−	−
Animal dander	+	+	−	−	+	−
Parietaria	+	−	−	−	+	+
Graminaceae	−	−	−	+	+	−
Ragweed	−	−	−	−	−	−
Olive	−	+	−	−	+	−
AR therapy:						
Oral Antihistamines	−	+	+	−	−	+
INCS	−	−	−	−	+	−
Oral Antihistamines + INCS	+	−	−	+	−	−
ADHD therapy						
Methylphenidate	−	+	−	−	−	+
Risperidone	−	−	−	+	−	−
Aripiprazole	+	−	−	−	−	−
Magnesium supplementation	−	−	+	−	+	−
TNSS scores	9	5	2	9	6	7
SNAP IV scores:						
Total	56	51	27	49	44	45
Inattention	21	22	11	21	20	10
Hyperactivity/impulsivity	15	13	9	13	14	19
Oppositional Defiant Disorder	20	16	7	15	10	16
PRQLQ/ARQLQ scores	119	51	19	98	74	67
MGL MAQ AR therapy scores	4	1	1	4	1	4
MGL MAQ ADHD therapy scores	4	4	1	2	1	4

+ yes, − no.

**Table 2 jpm-13-01346-t002:** Characteristics of the AR group.

Characteristics	Participants					
	1	2	3	4	5	6
Age	12	16	12	15	12	15
Natal preterm	−	−	−	−	+	−
Exposure to parental smoking	−	−	−	+	+	+
Neurological co-morbidities	+	−	−	−	−	−
Atopic co-morbidities	+	+	+	−	+	+
Positive SPT:						
House dust mites	+	+	−	−	+	−
Mold	−	−	−	−	−	−
Animal dander	−	−	−	−	−	−
Parietaria	+	−	+	+	+	−
Graminaceae	+	−	+	+	−	−
Ragweed	+	−	−	−	−	+
Olive	−	−	+	+	+	+
AR therapy:						
Oral Antihistamines	−	−	−	−	−	−
INCS	−	−	−	−	−	−
Oral Antihistamines + INCS	+	+	+	+	+	+
TNSS scores	6	5	6	4	7	6
SNAP IV scores:						
Total	59	4	4	1	0	0
Inattention	27	4	1	0	0	0
Hyperactivity/impulsivity	14	0	1	1	0	0
Oppositional Defiant Disorder	18	0	2	0	0	0
PRQLQ/ARQLQ scores	90	85	24	18	85	81
MGL MAQ AR therapy scores	1	4	4	4	4	4

+ yes, − no.

**Table 3 jpm-13-01346-t003:** Characteristics of the enrolled patients and TNSS, SNAPIV, and MGL MAQ scores in the AR-ADHD group and the AR group.

	AR-ADHD Group (*n* = 6)	AR Group (*n* = 6)	*p*-Value
Age, median (range)	14 (11–17)	13.5 (12–16)	0.89099 *
Natal preterm, n° (%)	1 (16.6)	1 (16.6)	0.77272 ç
Exposure to parental smoking, n° (%)	3 (50)	3 (50)	0.71645 ç
Neurological co-morbidities, n° (%)	4 (66.6)	1 (16.6)	0.12121 ç
Atopic co-morbidities, n° (%)	4 (66.6)	5 (83.3)	0.5 ç
Positive SPT, n° (%):			
House dust mites	6 (100)	3 (50)	0.09090 ç
Mold	1 (16.6)	0 (0)	0.5 ç
Animal dander	3 (50)	0 (0)	0.09090 ç
Parietaria	3 (50)	4 (66.6)	0.5 ç
Graminaceae	2 (33.3)	3 (66.6)	0.5 ç
Ragweed	0 (0)	2 (33.3)	0.22727 ç
Olive	2 (33.3)	4 (66.6)	0.28355 ç
AR therapy, n° (%):			
Oral Antihistamines	3 (50)	0 (0)	0.09090 ç
INCS	1 (16.6)	0 (0)	0.5 ç
Oral Antihistamines + INCS	2 (33.3)	6 (100)	0.03030 ç
TNSS, median (range)	6.5 (2–9)	6 (4–7)	0.57956 *
SNAP IV, median (range)	47 (27–56)	2.5 (0–59)	0.00841 *
MGL MAQ AR therapy, median (range)	2.5 (1–4)	4 (1–4)	0.25957 *

* *p* value by independent *t*-test. ç *p* value by Fisher’s exact test.

**Table 4 jpm-13-01346-t004:** PRQLQ scores (6–12 years) in the AR-ADHD group and the AR group.

	AR-ADHD Group	AR Group	*p*-Value
Total n° of patients 6–12 y/total n° (%)	2/6 (33.3)	3/6 (50)	
PRQLQ scores, median (range)	43 (19–67)	85 (24–90)	0.52716 *

* *p* value by independent *t*-test.

**Table 5 jpm-13-01346-t005:** ARQLQ scores (13–17 years) in the AR-ADHD group and the AR group.

	AR-ADHD Group	AR Group	*p*-Value
Total n° of patients 13–17 y/total n° (%)	4/6 (66.6)	3/6 (50)	
ARQLQ scores, median (range)	86 (51–119)	81 (18–85)	0.38089 *

* *p* value by independent *t*-test.

**Table 6 jpm-13-01346-t006:** Correlations between possible explanatory parameters and SNAP-IV subgroups scores in the AR-ADHD group.

	SNAP-IV, rPearson (*p*-Value)		
	Inattention	Hyperactivity/Impulsivity	Oppositional Defiant Disorder
Age, y	0.77957 (0.06757)	−0.17025 (0.74745)	0.21046 (0.68905)
TNSS	0.48156 (0.33356)	0.60948 (0.19905)	0.78597 (0.06385)
MAQ AR therapy	−0.03338 (0.95051)	0.61782 (0.1912)	0.70064 (0.12104)
MAQ ADHD therapy	0.12147 (0.81879)	0.64024 (0.17089)	0.87798 (0.02145)

**Table 7 jpm-13-01346-t007:** Correlations between possible explanatory parameters and SNAP-IV subgroups scores in the AR group.

	SNAP-IV, rPearson (*p*-Value)		
	Inattention	Hyperactivity/Impulsivity	Oppositional Defiant Disorder
Age, y	−0.37384 (0.46644)	−0.4561 (0.36340)	−0.49526 (0.31814)
TNSS	0.12035 (0.82042)	0.11581 (0.82707)	0.17857 (0.73509)
MAQ AR therapy	−0.98951 (0.00018)	−0.99612 (0.00002)	−0.99385 (0.00007)

**Table 8 jpm-13-01346-t008:** Analysis of possible explanatory parameters and AR therapy subgroups in the AR-ADHD group.

	Oral Antihistamines	INCS	Combination	*p*-Value by ANOVA
TNSS, mean	4.66	6	9	0.21464
SNAP-IV, mean	41	44	52.5	0.55655
MAQ AR therapy, mean	2	1	4	0.29629

**Table 9 jpm-13-01346-t009:** Correlation between n° of sensitizing allergens and TNSS in the AR-ADHD group and in the AR group.

	AR-ADHD Group	AR Group
TNSS vs. N° of sensitizing allergens, *p*-value *	0.01034	0.00051

* *p* value by independent *t*-test.

## Data Availability

The data presented in this study are available on request from the corresponding author.

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
