# Peer review of "Impact of the Allergic Therapeutic Adherence in Children with Allergic Rhinitis and ADHD: A Pilot Study"

_jpm, 2023, doi:10.3390/jpm13091346_

Round 1

Reviewer 1 Report

The article is written accurately and completely.

The purpose of this study is the effect of allergic treatment in children with allergic rhinitis and ADHD.  This study fills a gap in the field of mental health based on the relationship between allergic rhinitis and ADHD, and suggests that better adherence to AR treatment is associated with reduced inattention symptoms in children with ADHD.  The conclusions are consistent with the evidence and arguments presented and address the main question raised, and the references, figures and tables are appropriate.

Author Response

We are grateful to the Reviewer for the very positive comments related to our manuscript.

Reviewer 2 Report

The study is good, but the study sample size less.

Author Response

We are grateful to the Reviewer for the very positive comments related to our manuscript. As we underlined in the title and in the Materials and Methods, our study was designed as a pilot, monocenter, case-control study. The characteristics of the case group limited the sample size. Our purpose – as we wrote in the Conclusions – is to enlarge our sample in the future, with the idea of proposing a multicenter study.  

Reviewer 3 Report

This is a very interesting and pioneers pilot study. I will be waiting impatiently for the results coming from larger groups. Throughout the manuscript there are some minor spelling and syntax errors. They do not influence the overall high quality of the paper, however, I suggest that the manuscript is checked by English language editor before final publication. I recommend publication after the issues listed above are addressed and amended or rebuttal provided.  Please find my detailed comments and remarks in the attached file.

As in the attached report, minor issues mainly, minor editing required.

Author Response

“This is an interesting, innovative and inspiring pilot study regarding mutual associations of treatment adherence and AR & ADHD symptoms’ control – thanks for the opportunity to review it. The research has been well planned, the study groups, goals and methods are well described and clearly defined. The discussion is correctly carried out and major results are addressed in the discussion. Below, please find my remarks regarding this article.”

Response: We are grateful to the Reviewer for the very positive comments related to our manuscript.

Point 1: “In view of the fact that this is a pilot study including only 6 subjects in each group, I think it would be appropriate and more informative to add data regarding each subject, either in the scatter plot, or in the table. Also, the Authors are presenting the values as mean +/- SD. I think that in such small groups it would be better to present them as medians with ranges, unless the normal distribution was checked for.”

Response: Thank you for your suggestions. We integrated data regarding each participant in Table 1 and 2. Furthermore, we added medians with ranges in the other tables and in the manuscript.

Point 2: “In the MATERIALS AND METHODS section, please provide some more information about each questionnaire you have employed: TNSS, ARQLQ, PRQLQ, SNAP-IV, MGL MAQ, in particular point value ranges, interpretation and minimal clinically important difference (MCID)  - where applicable. You may expect readers from different fields of medicine (allergy, adolescent psychiatry, neurology, psychology), not everyone being familiar with tools applied in their area, therefore, short explanation can be of importance.”

Response: We added the information about each questionnaire used in the study, as you asked (lines 99-129). Our study design did not include the evaluation of MCID, but we think that it could be a good suggestion for the future larger study. Furthermore, We did not compare the efficacy of each AR treatment between the participants and every questionnaire was administered only once, without the possibility having a comparison of the effect of medication (as suggested by Meltzer et al, “Minimal Clinically Important Difference (MCID) in Allergic Rhinitis: Agency for Healthcare Research and Quality or Anchor-Based Thresholds?”, J Allergy Clin Immunol Pract. 2016 Jul-Aug;4(4):682-688.e6. doi: 10.1016/j.jaip.2016.02.006).

Point 3: “Lines 132-133: there is a considerable difference in treatment between AR-ADHD and AR groups, i.e. half of the ADHD subjects were on antihistamines only, whereas 100% of AR without ADHD were on intranasal GCS and antihistamine combination. DO the authors have any comments about the reasons for this discrepancy? Was this caused by different intensity of AR symptoms or had there been any other reason behind this?

Response: Differences in treatment between the two groups were principally due to the severity of the AR symptoms at the onset and the presence of atopic comorbidities (lines 168-169).

Point 4: “Lines 247-248: the Authors wrote that “contrary to expectations, patients with better ADHD therapeutic adherence showed higher AR symptoms”. This is quite intriguing finding and it would deserve some more comments. Could this be associated with differences in treatments between AR subjects with and without ADHD (see above)? Please provide some comment on that, if possible.”

and

Point 5: “Lines 251-252: it looks like ADHD subjects adhered better to the therapy of rhinitis, if this was a combination therapy. However, only half of the DHD subjects had been treated with nasal GCS plus antihistamine. Therefore, do the authors think that higher AR symptoms in spite of good adherence to therapy can be caused 50% of subjects by the therapy that did not include a basic anti-inflammatory agent (intranasal GCS)? I think there is some room for interesting speculations here.”

Response: Only 33.3% of AR-ADHD patients were under AR combination therapy, in contrast to the control group (100% in combination therapy). Data analysis showed that AR-ADHD patients with higher TNSS scores are those on dual therapy (INCS + oral anti-histamines) and with a more significant therapeutic adherence to both ADHD and AR medications [Table 8]. Due to the study's small sample, it is difficult to draw conclusions on the matter. One possible hypothesis could be that more AR-ADHD patients need combination therapy to achieve better control of the AR symptoms. Furthermore, we did not record TNSS at the onset of AR symptoms in our patients, so we cannot exclude the possibility that AR symptoms were worse at the diagnosis, and the decision to administer a combination therapy was consequential to the severity of the symptoms. It is also worth noting that all the patients in the AR-ADHD group had dust mite allergy (the principal cause of perennial allergic rhinitis) as opposed to only 50% in the AR group, in which SPT resulted negative for animal dander and moulds.

Point 6: “Overall, this is a very interesting and pioneers pilot study. I will be waiting impatiently for the results coming from larger groups. Throughout the manuscript there are some minor spelling and syntax errors. They do not influence the overall high quality of the paper, however, I suggest that the manuscript is checked by English language editor before final publication. I recommend publication after the issues listed above are addressed and amended or rebuttal provided.”

Response: We are pleased for your positive comments. We checked our paper for errors, as requested.
